# Integrating Text Mining into the Curation of Disease Maps

**DOI:** 10.3390/biom12091278

**Published:** 2022-09-10

**Authors:** Malte Voskamp, Liza Vinhoven, Frauke Stanke, Sylvia Hafkemeyer, Manuel Manfred Nietert

**Affiliations:** 1Department of Medical Bioinformatics, University Medical Center Göttingen, Goldschmidtstraße 1, 37077 Göttingen, Germany; 2Clinic for Pediatric Pneumology, Allergology and Neonatology, Hannover Medical School, Carl-Neuberg-Strasse 1, 30625 Hannover, Germany; 3Biomedical Research in Endstage and Obstructive Lung Disease Hannover (BREATH), the German Center for Lung Research, Carl-Neuberg-Strasse 1, 30625 Hannover, Germany; 4Mukoviszidose Institut gGmbH, In den Dauen 6, 53117 Bonn, Germany; 5CIDAS Campus Institute Data Science, Goldschmidtstraße 1, 37077 Göttingen, Germany

**Keywords:** text mining, disease maps, systems biology

## Abstract

An adequate visualization form is required to gain an overview and ultimately understand the complex and diverse biological mechanisms of diseases. Recently, disease maps have been introduced for this purpose. A disease map is defined as a systems biological map or model that combines metabolic, signaling, and physiological pathways to create a comprehensive overview of known disease mechanisms. With the increase in publications describing biological interactions, efforts in creating and curating comprehensive disease maps is growing accordingly. Therefore, new computational approaches are needed to reduce the time that manual curation takes. Test mining algorithms can be used to analyse the natural language of scientific publications. These types of algorithms can take humanly readable text passages and convert them into a more ordered, machine-usable data structure. To support the creation of disease maps by text mining, we developed an interactive, user-friendly disease map viewer. The disease map viewer displays text mining results in a systems biology map, where the user can review them and either validate or reject identified interactions. Ultimately, the viewer brings together the time-saving advantages of text mining with the accuracy of manual data curation.

## 1. Introduction

Every day, more and more data and knowledge on different diseases and their underlying biological pathways are being acquired. Thus, it is becoming increasingly important to develop methods of data and knowledge integration, storage, and representation in ways that can be interpreted and analysed by humans and computers alike. One of these approaches is systems medicine disease maps, which has been proposed by Mazein et al. in 2018. The authors define disease maps as a “comprehensive, knowledge-based representation of disease mechanisms” [1]. They evolved from and are comparable to metabolic and signaling pathways, stored and represented in standardized formats such as the Systems Biology Graphical Notation (SBGN) [2] or Systems Biology Markup Language (SBML) [3]. A major difference between metabolic or signaling pathways and disease maps is that the latter are not limited to biochemical or regulatory relations between entities but can also include physiological ones. Disease maps can be used for a multitude of purposes, such as identifying disease biomarkers and drug targets, drug repositioning, structuring omics data, and developing improved diagnostics [1,4]. Most recently, a large, interdisciplinary community of over 230 researchers launched a project to create a COVID-19 disease map [5]. This resulted in what, to the best of our knowledge, is the largest disease map to date, currently consisting of 5499 elements, which are connected by 1836 interactions across 42 diagrams. The data for this enormous knowledge resource were curated from 617 publications and preprints, highlighting the sheer time and manpower required to create these manually curated disease maps. One way to support the construction of disease maps is text mining, the automated annotation of texts that produces a condensed keyword list, which can then be formatted into machine- and human-readable media and to consist of the core information of that text. In principle, text mining means the extraction of information from textual data, thereby speeding up the curation and annotation process of human-written text [6]. To do so, many possible information technologies are applicable, for example, machine learning, pattern matching, or the processing of natural, human-readable language [7].

In general, a text mining algorithm will follow the steps below. As an input, the algorithm will take a human-readable sentence, in this case from a biological paper. It will then first highlight the named entities (NE), which are terms that are then normalized and transformed into identifiers. These NEs can be proteins, genes, diseases, or any other biologically relevant term, taken from an underlying database that contains NEs that the system should be able to identify. This recognition (named entity recognition (NER)) is crucial for the success and effectiveness of text mining and is therefore a focus of refinement to increase the specificity and sensitivity of the algorithm. The entities are then assigned to unique identifiers, which are then organized into an identifier scheme. Afterward, the extracted relationships from the input text data are included between named entities. The resulting network of nodes and relationships can then be compared and expanded with additional text data. With the help of this network, new hypotheses can be formed and these can then be the subject of further research [7].

One of the main challenges in NER is the multitude of different identifiers for almost anything in biology or chemistry, sometimes varying greatly between different publications and databases. This variation in names for the same biological entity needs to be recognized and normalized by the algorithms. In addition to these intended differences in nomenclature, there are more variations that need normalization: for example, variations in orthography (“amino acid” vs. “amino-acid”), abbreviations, and spelling variations, such as upper/lower case or American vs. British English wording [8]. All these variations must be taken into account, and the term needs to be assigned to the same biological identifier, which then results in a list of possible terms all referring to the same ID. When it comes to interactions, even more words can be used to describe similar relations between entities. The system needs to recognize the buzzwords for relations and the entity terms to create the desired entity-relationship model. Moreover, differences in the structure of the sentence in combination with the wording can be challenging to the system.

Text mining has been gaining more and more applications in scientific projects over the last two decades. The principle and technique of data mining have been known since the late 1990s but have not been widely used by the scientific community [9]. In particular, in systems biology and biomedicine, the use of text mining can be of essential value. Even if those scientific fields rely heavily on data stored in unified formats and databases to ensure cross-author usability, a substantial proportion of essential information is still only available as text in human-written publications. As of now, there are many algorithms that are specialized for biological terms that are implemented as NER. In order to establish, compare, and evaluate common standards challenges such as BioCreative (http://www.biocreative.org, accessed on 1 September 2022) have been put into place, which aims to compare methods and critically assess scientific progress in text mining [10]. Currently, biological text mining and NER specifically already find applications in the curation of different databases. For example, the BRENDA database (BRaunschweigENzymeDatabase; http://www.brenda-enzymes.org, accessed on 1 September 2022) [11], which collects enzyme functional data, employs text mining approaches to extract kinetic data from PubMed abstracts [11]. Furthermore, the protein interaction database STRING (https://string-db.org/, accessed on 1 September 2022) uses text-mined data to identify protein–protein interactions [12]. A more extensive overview of many more examples of existing text mining applications with a focus on cancer research can be found in Zhu et al., 2013 [7].

Nonetheless, even though great strides have been made in the development of text mining algorithms with high sensitivity and specificity, they cannot yet replace a human expert curator. We, therefore, developed a tool to bring together the advantages of text mining and the expert knowledge and experience of scientists to support the creation of systems biology disease maps. Our tool consists of an interactive disease map viewer, which takes the output of an independent text mining system, translates it to the required format, and displays it in a disease map-like cellular layout. This allows the user to utilize the text mining approach they find most suitable for their use case or even include results from more than one system. The user then has the possibility to examine the interactions identified by the text mining algorithm and evaluate them based on the text passage they are based on. In the end, this results in a list of automatically parsed but expert-validated interactions, which can then be used as a basis for a disease map. Ultimately, this simplifies and significantly speeds up the curation step during the construction of disease maps.

## 2. Materials and Methods

### 2.1. Data Preparation

To bring text mining results into a format appropriate for further use, the results from an independent text mining algorithm were brought into a simple, reproducible format, consisting of two tables in CSV format. One table consists of all mined entities and their subcellular localization, and the other includes derived interactions observed between them. The tables were then parsed into the JavaScript Object Notation (JSON) format. The JSON format is a very storage-efficient way to save and interchange data between different JavaScript applications. Currently, it is widely used for providing data to the user from a server or a web service, where data can be parsed via a host’s API (Application Programming Interface). The conversion of the tables into JSON format was performed using Python with the libraries: pandas [13,14], numpy [15], libsbgnpy [16], and json.

The resulting JSON file was then further used as an exemplary disease map for the disease map viewer.

### 2.2. Disease Map Viewer Implementation

The disease map viewer tool was implemented with the Cytoscape.js library [17]. Cytoscape.js is the JavaScript variant of the Cytoscape software [18]. Cytoscape itself is an open-source project for accessing and viewing graphical networks inside a downloadable instance. This software can be programmatically accessed and therefore personalized and implemented into our tool via the JS library Cytoscape.js. Another big advantage of the Cytoscape.js variant is the capability of loading data dynamically while the user is browsing the graphical map. Furthermore, it is possible to load big maps in a memory-efficient manner into the Cytoscape.js instance using the JSON format.

To make our Cytoscape.js instance accessible, we used Grails and our previously developed CandActBase [19] as an underlying framework. We used the AJAX (Asynchronous JavaScript and XML) protocol to dynamically load data into a JavaScript application from a web server [20]. AJAX is capable of loading data dynamically based on the input of the user, even if the website has already been loaded completely. The AJAX call will access a defined URL (in this case, a local file) and load the data into the JS script. This data can then be processed, altered, and presented by the rest of the code. This asynchronous behaviour makes AJAX valuable for our purpose and improves the speed of the script significantly.

## 3. Results

In order to support the creation of disease maps, we developed a tool capable of displaying text mining results as disease maps and validating them through the integration of expert domain knowledge.

For this purpose, we used an independent, exchangeable text mining algorithm to parse molecular interactions between biological entities’ data from publicly available scientific text. The results are output in two simple, reproducible CSV files, one containing the interactions between the entities themselves and the other specifying their subcellular localization. A flowchart of the input data, software, and output data of the systems can be seen in Figure 1.

To prepare text mining results that are easy to store, share, and use, we used a Python script to convert them from a simple CSV file to JSON format. Simply put, the JSON data structure of the text mining results is a list of every element (nodes, compartments, and edges) in the disease map. Depending on the element, the structure differs slightly. Each element has three basic properties: “data”, “position”, and “group”. The “group” specifies if the element is a node or an edge, i.e., a molecular entity or an interaction. The “position” property, which is automatically created by the python script, sets the x and y parameters to assign it to a specific location on the map. The most advanced property of each element is the “data” property, where all associated data are stored. Additionally, edge-type entities have the property “classes”, where the category of the interaction is defined (“neutral”, “inhibit”, “activate”, and “undefined”). Further properties are the unique identifier, and cytoscape.js-specific parameters (For more external information visit https://js.cytoscape.org/, accessed on 1 September 2022). The following additional parameters are important for representation in the SBGN format: For nodes, the “label” is the name specified, and the “parent” is the cellular compartment in which the gene is active. For edges, the start and end nodes are defined by the respective identifiers. Furthermore, all edges have a parameter called “references”, which lists the PubMed IDs of all references this edge is based on. For each reference, the PubMed ID is given together with the sentence where the interaction was identified. Moreover, all verbs found in those sentences as well as the categorization of those verbs are stored.

This SBML-based JSON format is used by the Cytoscape.js library to create the graphical SBGN map from it.

The interface is built around the Cytoscape.js instance that renders and displays disease maps to help the user annotate and review the text-mined disease map conveniently.

Figure 2 shows the interface with exemplary data. The main graph is shown in a cell-like layout, where the user can zoom in and out. The rectangular nodes represent the molecular entities and are localized in the subcellular compartment specified in the JSON file. The arrow-shaped edges represent molecular interactions between them. All entities (genes/proteins and compartments), as well as their respective edges, can be moved freely by dragging to improve structure and visibility to fit the user’s needs.

The colouring is the colour of categorization of found verbs. All “activating” edges are coloured green, “inhibiting” edges are coloured red, “neutral” edges are coloured blue, and “undefined” edges have a grey colour, while incoherent interactions are shown in brown.

The left sidebar shows the legend and filter options for the edges in the graph. As a default, all edges are displayed, but the user can uncheck types of edges to hide them and thus obtain a better overview of the remaining categories of edges. This legend can be opened and closed by clicking the top button “hide/show filter”.

Another way the data from the text mining are categorized is by the thickness of the edges in the graph. The more distinct publications have been found to have both connected nodes mentioned in the same sentence, the thicker the edge between them. In the bottom-left corner of the filter window, the user can filter the edges depending on the number of supporting publications. The slider can be moved to define a minimum number of publications an edge needs to have to display it. Moreover, below the slider is a button that will reset the filter and reload the map.

Another feature of the disease map/SBGN map viewer is the timeline function. As an interesting use case of our text mining workflow, we chose to create a timeline made from SBGN maps from publications published in different years and, thus, show the focus of research in the past. To obtain biological interactions that are associated with the query subject over time, we categorized texts by their years of publication. Thus, we created exemplary momentary snapshots over the years. The user can choose which disease map from which year they would like to access between the years 1990 and 2020 in 5-year steps.

In order to integrate expert knowledge and validate text-mined data, we included a review function, as observed in the right-hand panel of the interface. The user can examine all interactions with two methods: by clicking the “Next edge” button to iterate all interactions that need to be reviewed or by directly selecting a specific edge from the graph. The review panel will then display the two nodes connected by the clicked edge and the colour of the edge between both, as well as the current review status of the interaction. Below this, a list of PubMed IDs is displayed together with the sentences that have been used to identify the interaction in each reference. The verbs that have been used to categorize the interaction are coloured in red. The user can then load the entire text to obtain more context for the sentence. The user can then review the interaction with all available data on hand and assign a status to the interaction. If the expert approves the text-mined interaction, the “accept” status can be selected. If the text-mined interaction is a false positive, the “decline” status is appropriate, and if more research needs to be conducted to approve the interaction, the “further inspection needed” status can be assigned.

To view the status of the review process, the data can be downloaded either as a CSV file with all interactions, their current review status, and the PubMed ID from with the interaction, which was text mined from the disease map, or as a JSON file with the entire disease map in a JSON object that can be saved for reloading in a later session or to share with other users.

## 4. Discussion

With more and more biological and biomedical data being published, more knowledge is available and needs to be processed and structured. One way to do so in biomedicine is by using disease maps that visualize and describe disease pathways in a human- and machine-readable medium [1]. However, with the increasing number of publications every year, new computational approaches are needed to support researchers and clinicians in filtering and annotating large data sets to extract scientifically meaningful knowledge. Here, we propose a tool to (re)view text-mined data and display it appropriately to accelerate the curation process of textual data significantly. It spares the researcher from having to manually read large sets of publications to construct or curate disease maps but allows them to conveniently iterate text-mined interactions and preprocessed publications to verify found interactions.

Our tool can be used in combination with text mining software to preprocess large textual data sets and review the text mining results easily to ultimately combine the advantages of the speed of text mining with the accuracy of manual data curation reviewed by experts in the scientific field in question. The interface is kept clearly laid out and is easy to use; thus, researchers with limited experience in computational software can use it intuitively.

To ensure maximum transparency, the text-mined data can be reviewed in a very detailed manner. Every text-mined interaction can be examined to see which terms in which sentences from which publications were used to identify an interaction. In this way, the reviewer can closely inspect if the interaction is a true or false positive and mark the interaction accordingly. Moreover, all data can be downloaded at every step of the curation process. In this way, the data can be shared with co-workers and peer-reviewed easily. For this purpose, standardized data formats are used to ensure the exchangeability of the input text mining data. Therefore, the viewer can use interchangeable external text mining software just by making little adjustments to the input data. This is important with respect to the rapid improvement of text mining algorithms. The tool can be used to display results from all kinds of text mining software and can be employed for comparison purposes.

To the best of our knowledge, this is the first tool that integrates text mining directly into the disease map curation process. Several different tools have been developed to extract interactions between biological entities and can create protein–protein Interaction (PPI) networks [21,22,23]. The HPIminer, for example, uses NER to identify interactions from sentences and then adds information from PPI databases and additionally extracts, overlays, and displays KEGG (http://www.kegg.jp, accessed on 1 September 2022) pathways from the two interacting proteins [21]. All these tools come with their own highly sophisticated text mining algorithms and include different data sources to produce extensive networks. In contrast, our tool does not focus on text mining itself but on making the results from the already existing, high-quality text mining tools usable and integratable. Users can use their preferred text mining tool or algorithm and visualize the results in the disease map viewer so domain experts can verify the data and then further utilize it, e.g., in a disease map.

To show how the viewer operates, we used an individualized text mining workflow to create a sample data set with the use case of cystic fibrosis, based on the CFTR Lifecycle Map we previously curated [24].

The disease map viewer, installation instructions, and the exemplary cystic fibrosis data set are available under https://s.gwdg.de/8bK6f5, accessed on 2 September 2022 (Appendix A).

## 5. Conclusions

We developed a tool to create an interface between biological text mining and the creation of systems medicine disease maps. Our disease map viewer takes the interaction data extracted by a text mining algorithm of choice and displays it in a cellular layout and interactive manner. Domain experts can then intuitively examine individual interactions and validate or reject them, and the verified interactions can be exported for further use. This supports the creation of disease maps and systems biological models, as it brings together the speed of automated text mining and the high accuracy of human expert knowledge, thereby using the benefits of both without sacrificing quality or time effectiveness.

## Figures and Tables

**Figure 1 biomolecules-12-01278-f001:**
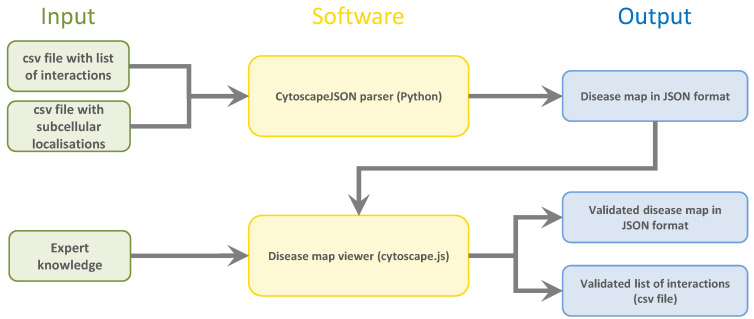
Flowchart of the processes included in the tool. Input knowledge and data are shown in green on the right, the software modules are shown in yellow, and the output files are shown in blue on the right. Two CSV files, one containing the list of interactions and one containing the subcellular localisation of the entities, serve as input for the CytoscapeJSON parser implemented in Python. The resulting JSON file serves as input for the disease map viewer, where the interactions are validated by expert knowledge. The validated interactions can then be exported in a cellular layout in a JSON file or as a list of interactions in a CSV file.

**Figure 2 biomolecules-12-01278-f002:**
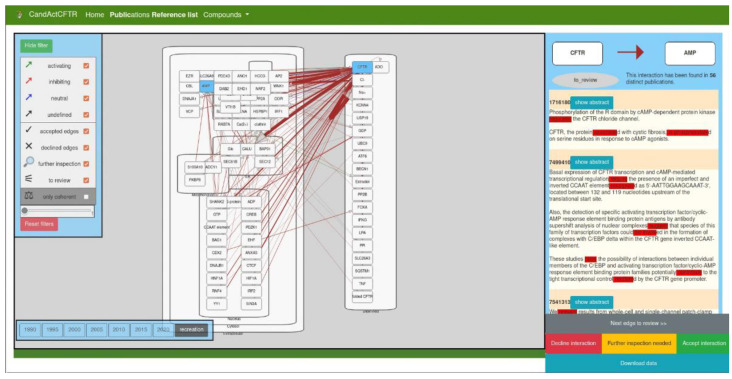
Interface of the disease map viewer. The large window in the middle shows the text mining data as a coarse disease map in a cellular layout. The left sidebar shows the legend and filter options, and the right sidebar shows the review function, where the supporting sentences from the parsed publications are displayed and the user can validate or reject an interaction. The buttons on the bottom left show the timeline option, where the interaction data can be filtered by date of publication.

## Data Availability

Not applicable.

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
