# Peer review of "Integrating Text Mining into the Curation of Disease Maps"

_biomolecules, 2022, doi:10.3390/biom12091278_

Round 1
Reviewer 1 Report
The authors have developed a viewer that integrates text mining to support the creation of systems biology disease maps. This is an important technical study that can help biologists or clinicians to interpret sizeable biomedical data sets and to be processed and structured. However, I would suggest some necessary revisions as follows:
1- In the introduction (Lines 92-105), I would ask the author to go through the literature and rewrite this section to help lay the stage for the rest of the paper.
2- In the introduction (Figure 1), the authors copied most of figure 2 from Harmston et al., 2010 and even used the same biological sentence. The authors should either delete figure 1 or include their original formation to illustrate the system workflow and move it to the Materials and Methods section.
3- I suggest improving the quality of figure 2 to increase the readability.
4- In the discussion, the authors mention the advantages of their system and never cite other available systems for comparison purposes.
5- Please add a conclusions section.
6- I would ask the authors to write the installation instructions in the README.md file in a way that biologists or clinicians follow and understand as they are the end users of their system.
Author Response
The authors have developed a viewer that integrates text mining to support the creation of systems biology disease maps. This is an important technical study that can help biologists or clinicians to interpret sizeable biomedical data sets and to be processed and structured.
We thank the reviewer for their kind assessment of our tool and for taking the time to review our manuscript.
1- In the introduction (Lines 92-105), I would ask the author to go through the literature and rewrite this section to help lay the stage for the rest of the paper.
We thank the reviewer for pointing out the issues with this paragraph. We have rewritten it and describe the examples in more detail so they set the frame for the rest of the manuscript.
2- In the introduction (Figure 1), the authors copied most of figure 2 from Harmston et al., 2010 and even used the same biological sentence. The authors should either delete figure 1 or include their original formation to illustrate the system workflow and move it to the Materials and Methods section.
We thank the reviewer for their suggestion and have decided to remove the figure. We kept the description of the general steps in text mining in the introduction, as we think it will be useful for readers not entirely familiar with the text mining process.
3- I suggest improving the quality of figure 2 to increase the readability.
We thank the reviewer for pointing out the issues with the figure quality and have exchanged it. As it is a screenshot of the interface, the resolution is unfortunately limited, but its readability is now improved. Additionally, there is separate PDF file with a larger version of the figure, which, although still pixelated when zooming in, is better to read than the one embedded in the text due to its larger size.
4- In the discussion, the authors mention the advantages of their system and never cite other available systems for comparison purposes.
We have added a paragraph to compare our tool to other systems. To the best of our knowledge, our tool is the first of its kind, since most systems in the field develop novel text mining algorithms and tools. Our tool does not aim at the text mining process itself, but rather at creating an interface between already existing, highly effective text mining algorithms and manual quality control through experts to support the curation of disease maps. After extensive literature we could not find a similar tool, but would be grateful for references if he reviewer knows of one. The new paragraph now cites publications with other systems and states the difference more clearly.
5- Please add a conclusions section.
We thank the reviewer for their suggestion and have included a conclusions section to briefly sum up the main points.
6- I would ask the authors to write the installation instructions in the README.md file in a way that biologists or clinicians follow and understand as they are the end users of their system.
We thank the reviewer for their pointing this out and have rewritten the documentation accordingly. It is now more detailed and shows screenshots and examples for the different steps.
Reviewer 2 Report
Could you describe in the manuscript how potential users can test your tool?
Additionally, could you show more case study examples? You could include prints and more details in the supplementary material.
Minors:
Line 1: Please, include the main link in the abstract.
Line 23: change “type” to “types”.
Line 25: change “user friendly” to “user-friendly”
Line 45: Please, correct the apostrophe near: ‘omics-data
Lines 140 and 142: Please, standardize the way of citing the Cytoscape.js tool. You used “Cytoscape.JS” (line 140) and “cytoscape.js” (line 142).
Line 248: in the sentence end, change “,” to “.”
Author Response
Could you describe in the manuscript how potential users can test your tool?
Additionally, could you show more case study examples? You could include prints and more details in the supplementary material.
We thank the reviewer for their suggestions. We have updated the documentation in GitLab, where we also included descriptions of our sample data. There, we show detailed instructions in how to use the tool together with screenshots for better visualization. We think the documentation is the best place for these instructions, as they can be updated easily when required, and hope the reviewer agrees.
Minors:
Line 1: Please, include the main link in the abstract.
Line 23: change “type” to “types”.
Line 25: change “user friendly” to “user-friendly”
Line 45: Please, correct the apostrophe near: ‘omics-data
Lines 140 and 142: Please, standardize the way of citing the Cytoscape.js tool. You used “Cytoscape.JS” (line 140) and “cytoscape.js” (line 142).
Line 248: in the sentence end, change “,” to “.”
We thank the reviewer for pointing out these errors. We corrected them in the revised manuscript.
Reviewer 3 Report
-explanation is needed on: “which takes the output of in principle every text mining algorithm ..” for the meaning of “ every text mining algorithm” an example would be helpful, line:111
-to facilitate understanding of the various operations of the proposed tool, a flowchart is needed that will include the input(s) of the tool, the different kinds of processes (with their outputs), and the final output of the tool.
-a step by step example with data of the tool operations would be useful
Author Response
-explanation is needed on: “which takes the output of in principle every text mining algorithm ..” for the meaning of “ every text mining algorithm” an example would be helpful, line:111
We thank the reviewer for pointing this out. We have rewritten the sentence and explained it better. As our tool is a stand-alone program and does not perform text mining itself, it is completely independent of the text mining algorithm used to produce the input data. Users can use their own text mining systems or ones already published by others and input the results into our tool.
-to facilitate understanding of the various operations of the proposed tool, a flowchart is needed that will include the input(s) of the tool, the different kinds of processes (with their outputs), and the final output of the tool.
We thank the reviewer for the suggestion and added a flow chart describing the input, software modules and output of the system.
-a step by step example with data of the tool operations would be useful
We thank the reviewer for the suggestion. We have updated the documentation of the tool in GitLab to now include detailed instructions and descriptions of the tool’s functionalities together with screenshots for better visualization. We hope the reviewer agrees, that the documentation is the most suitable place for these examples and instructions, as they can be updated easily when required.
Round 2
Reviewer 1 Report
Thank you for addressing my concerns and comments. The current version of the manuscript has much improved and will interest the journal's readership.
Reviewer 2 Report
The authors improved the documentation and included more details about the installation process.